

# Planning for climate change impacts on hydropower in the Far North

Jessica E. Cherry[1], Corrie Knapp[2], Sarah Trainor[3], Andrea J. Ray[4], Molly Tedesche[5], Susan Walker[6]

[1]International Arctic Research Center and Institute of Northern Engineering, University of Alaska Fairbanks, Fairbanks, Alaska 99775, USA
[2]Department of Environment & Sustainability, Western State Colorado University, Gunnison, Colorado 81231, USA
[3]Alaska Center for Climate Assessment and Policy, University of Alaska Fairbanks, Fairbanks, Alaska 99775, USA
[4]Earth System Research Laboratory-Physical Sciences Division, National Oceanographic and Atmospheric Administration, Boulder, Colorado 80305, USA
[5]International Arctic Research Center, University of Alaska Fairbanks, Fairbanks, Alaska 99775, USA
[6]National Oceanographic and Atmospheric Administration-National Marine Fisheries Service, Juneau, Alaska 99801, USA

*Correspondence to*: Jessica E. Cherry (Jessica.Cherry@alaska.edu)

**Abstract.** Unlike much of the contiguous United States, new hydropower development continues in the Far North, where climate models project precipitation will likely increase over the next century. Regional complexities in the Arctic and sub-Arctic, such as glacier recession and permafrost thaw, however, introduce uncertainties about the hydrologic responses to climate change that impact water resource management. This work reviews hydroclimate changes in the Far North and their impacts on hydropower; it provides a template for application of current techniques for prediction and estimating uncertainty, and it describes best practices for integrating science into management and decision-making. The growing number of hydrologic impacts studies suggests that information resulting from climate change science has matured enough that it can and should be integrated into hydropower scoping, design, and management. Continuing to ignore the best-available information in lieu of status quo planning is likely to prove costly to society in the long term.

**Keywords:** hydropower; hydroelectric power; climate change impacts; water resources; Arctic climate change impacts; Alaska; best practices in climate policy; Susitna River; Susitna-Watana Dam

## 1 Introduction

Generation of conventional hydroelectric power in the contiguous United States has declined over the past several decades, while use of other renewables has grown considerably (Fig. 1). Hydropower generation has declined due to the impacts of precipitation variability and drought, enforcement of in-stream flow regulations, and a drop in capacity when dams are removed (EIA, 2015; Malewitz, 2014). Few new facilities have been built in the U.S. because of the impacts of dams on fish





habitat, challenges of property and water rights, the high cost of dam construction, and simply because easy-to-develop projects were already built in the early years of the national reclamation movement (Gallucci, 2014; Malewitz, 2014). In contrast, both hydropower generation and capacity in the Far North--Alaska, Canada, and much of Scandinavia—are actively growing (Figs. 2 and 3). In Alaska, in particular, several new projects are being proposed, including the massive 2.8 million

MWh Susitna-Watana Dam (AEA, 2016; REAP, 2016).

While the Far North (defined here as the Arctic and sub-Arctic) may have considerable water resources (e.g. AEDI, 2016; Conner and Francfort, 1997), questions about climate change impacts on existing and proposed hydropower projects relate to unique environmental complexities that require consideration across the boreal, sub-Arctic, and Arctic regions. How will

climate change impact glacier runoff? How does thawing of permafrost (soils that used to remain frozen throughout the year) impact partitioning of runoff into surface and subsurface waters? Where and when will the phase of precipitation change from snow to rain and how will total precipitation volumes and timing of storage change? How do changes in air and water temperatures impact the timing of snowmelt, river ice cover, and runoff? Do changes in glaciers and ground ice impact river sediment loads and subsequently fish habitat and potential or existing hydropower infrastructure? While these climate-

related questions also apply to alpine hydropower areas in the contiguous United States, they are major concerns for hydropower in the Far North and have received little attention. Figure 4 shows that the percent of hydropower capacity used in the Far North has actually decreased in several countries, suggesting either the impacts of a changing climate, or a misalignment of infrastructure and resources, or both. Because of the large proposed project on the Susitna River in central Alaska, and ongoing development throughout the Far Northern region, there is an urgent need to understand these issues.

There is widespread consensus that climate change will impact water resources globally and regionally (Georgakakos et al., 2014; Jiménez Cisneros et al., 2014; Mukheibir, 2013; Beniston, 2012; Viviroli et al., 2011; Fenner, 2009; Vicuna et al., 2008; Barnett et al., 2005; Payne et al., 2004). The volume and timing of runoff in many northern locations are already changing (e.g., Wolken et al., 2015; Dahlke et al., 2012; Jones & Rinehart 2010, Wilson et al., 2010), as well as the amount

and form of precipitation (Walsh et al., 2014; Callaghan et al. 2011a,b). Increasing precipitation patterns are expected to favor northern latitudes (Fig. 5), increasing the water available for hydropower production (Jiménez Cisneros et al., 2014; Hamududu & Killingtveit 2012). However, the impacts of climate change will be further complicated by several factors including glacial wasting, permafrost thaw, and erosion (Bliss et al., 2014; Quinton et al. 2011; Wada et al. 2011). These characteristics of northern systems may impact the long-term risk associated with hydropower projects through changes to

in-stream flow, increased sedimentation, and vulnerability to hazards such as glacial burst flooding and seismic activity related to changes in water/glacier mass loading.

Because of the long design life of hydropower infrastructure (typically fifty to one hundred years or longer), it is important to consider climate change and impacts unique to Far Northern regions in both the project planning and operations phases





(Viers, 2011; Markoff and Cullen, 2008). Uncertainties related to these climate impacts may affect the viability of a proposed project. The intent of this article is to discuss the state of the relevant climate science with a focus on Far Northern regions, to describe methods for predicting water supply and estimating uncertainty in northern hydroclimatology, and strategies for integration of climate change science into hydropower management and barriers to this process. Finally, we

propose best practices for the use of climate change information in planning new projects and operating existing facilities, to incorporate the range of possible futures represented by uncertainty.

## 2 Climate change impacts on hydropower

### 2.1 Scope of Study

The scope of this article is to describe the observed and projected climate impacts that would affect proposed and existing projects and how this information relates to planning and management. In the subsections below, we will discuss general impacts of climate change and variability *on* hydropower systems followed by a focus on Far Northern regions. We acknowledge that hydropower projects may be an opportunity to *mitigate* global climate change, and in some regions may represent the best possible power source for minimizing the carbon footprint, environmental pollution, and feedbacks to the

global climate. Interest in the sustainability of hydropower relative to other sources of energy is contributing to its growth trend in regions outside of the contiguous U.S. and, as we will argue, hydropower infrastructure is vulnerable to climate change and variability.  This growth in hydropower development makes it critical to study the linkages between hydrologic change and project risk (Harrison et al., 2003).

Likewise, there are significant effects that hydropower projects have on their environs, which are arguments against their sustainability. These include destruction of fish habitat and damage to the ecosystems that depend on them. They include flooding of land that previously had other ecosystem functions and may have contained private property or cultural resources. There are also changes to local microclimates and enhanced evaporation caused by reservoirs in conventional, large hydropower projects. These feedbacks *from* hydropower facilities onto the environment, again, are not the focus of this

article. In the subsections below, we will discuss general impacts of climate change and variability *on* hydropower systems followed by a focus on Far Northern regions.

### 2.2 Climate change impacts on hydropower supply and demand

The water supply for generating hydropower is derived from precipitation (rain and snow), surface water, groundwater that

makes its way to the surface, and glacial melt in northern and alpine areas. In much of the tropical and temperate regions, water supply is a balance of precipitation, evapotranspiration (ET), and storage. Globally, trends in precipitation have been



difficult to detect, but the IPCC AR5 described the increase in historical precipitation from 60-90° N as *likely*[1]. The problem of trend detection in the Far North is more challenging than in warm climates because solid precipitation is particularly difficult to measure and much of the year, precipitation falls as snow. Measurement errors, siting biases, poor spatial coverage, and network heterogeneity over time all reduce the confidence in trends of snow (Cherry et al., 2005a, 2007).

However, for three out of four global datasets, the trend for precipitation between 60-90° N was shown by the IPCC AR5 authors to have increased for the period 1951-2008, and more so than in any other region.

McAfee et al. (2013, 2014) published two studies that focused on estimating precipitation trends in Alaska. Using station data and gridded analyses, these studies showed that inhomogeneity in the observational networks makes trend detection

here nearly impossible at this time. Those authors found that virtually no significant trends in precipitation could be derived directly from station data. According to McAfee et al. (2013, 2014), the gridded Global Precipitation Climatology Center (GPCC) data set appears most reliable, shows an increase in annual precipitation in Arctic Alaska, and a slight decrease in annual precipitation in the southern half of the state between 1980 and 2008, though few regions show any statistical significance. By season, the pattern is much the same.  For the longer period of 1950-2008, all but Northwest Alaska in

winter appears to be drying or largely unchanged (McAfee et al., 2013, 2014).  These authors also reviewed more than a dozen other studies on precipitation trend analyses in Alaska and showed that results are entirely dependent on the sites used and the period of time selected. Rawlins et al. (2010) did a similar calculation for precipitation trends across the pan-Arctic using observations, climate models, and reanalysis data, and found increasing precipitation trends from 1950-2008 and 1980-2008 for all but one data set.

Of these precipitation trend analyses, few have focused on long-term snowfall in particular. In Canada, however, a study found that snowfall increased in the northern part of the country, but actually decreased in the southern part (Mekis and Vincent, 2011).  Suffice it to say that into the future, this will most likely be the pattern, from first principles: because of the warming atmosphere and enhanced transport of moisture from lower latitudes, the northern-most regions of the Far North

will likely receive more snowfall and the southern portion of the Far North will likely receive more rain in a changing climate. These results are also seen in modeling studies (e.g. Peacock, 2012). For hydropower in the Far North, this is likely to mean more supply available overall. In the southern-most parts of this region, more precipitation will arrive in liquid form and be available earlier in the season for hydropower generation, rather than stored in the snowpack until spring melt. In the Far North, ET is a relatively small portion of the hydrologic budget and precipitation dominates (Kane et al., 1990), though

ET will increase as the growing season continues to lengthen as it has in recent decades (Genet et al., 2013; Olchev and Novenko, 2011; Zeng et al., 2011). Other aspects of the supply side, which include groundwater storage and glacier storage, will be addressed in the next section on regional complexity.

---

[1] The IPCC defines *likely* as having a 66-100% probability (Mastrandrea et al., 2010).





Air temperature increases in the Far North are a robust signal and the IPCC AR5 report denotes *high confidence[2]* in these trends. In the Arctic, temperature has had the biggest increasing trend in autumn (September-November) over the past three decades (Cohen et al., 2012a; Alaska Climate Research Center, 2015). This increase in autumn air temperatures may delay the establishment of the snowpack and reduce availability of water in reservoirs during the following spring. An increase in

the number of rain on snow events, driven by warming, can also lead to more available water in northern reservoirs during the cold season. Air temperature trends also have a major impact on hydropower *demand*, especially in areas where home and commercial heating is supplied by this electricity. Many places in the Far North have warmed considerably during the winter season, but in other places, recent modeling work has shown that colder winters, particularly in northern Europe over the last three decades, are also consistent with climate change (Cohen et al., 2012b). Seasonal asymmetries in warming

trends (Cohen et al., 2012a) certainly have the ability to change the timing and magnitude of the demand for hydropower.

The co-variability of temperature and precipitation anomalies is interesting to study because it can affect the impact or hardship experienced by hydropower users. For example, if there is a shortage of precipitation, but a warm air temperature anomaly, the impact of the water shortage might be much less than if there were a wintertime cold anomaly. Cherry et al.

(2005b) showed that the co-variability of Scandinavian precipitation and temperature driven by the North Atlantic Oscillation (NAO) trends tended to cause significant impacts because cold winters occurred during dry anomalies and the hydropower supply was short when demand was high. Modes of climate variability such as El Nino-Southern Oscillation (ENSO) and the NAO influence temperature/precipitation co-variability as well as seasonal asymmetries in long-term climate trends.

For both precipitation and temperature, there is some debate whether non-stationarity (change in the long-term mean) persists over the period of instrumental record (Rhines and Huybers, 2013; Hansen et al., 2012) and this has obvious implications for reservoir construction and management (Brekke et al., 2009). However, changes in climate variability, including location, timing, magnitude, and extreme events also have implications for reservoir construction (i.e. design

capacity) and management (i.e. timing of generation, flood control, and water spilling).

There are relatively few publications about changes in extreme events for the Far North, probably due to the same issues with data quality that affect our confidence in basic trends. However, atmospheric circulation has shifted poleward since the 1970s and this has pushed the storm tracks northward and contracted the polar vortex (Hartmann et al., 2013), providing a

physical basis for changes in extremes at a given place. An assessment by Melillo et al. (2014) shows a significant decrease in cold days and cold nights and a significant increase in warm days and warm nights, observed for the high latitudes since 1950. Bennett and Walsh (2015) looked at historical and projected changes in extreme temperature and precipitation events

---

[2] The IPCC defines *high confidence* as having both high agreement and robust evidence (Mastrandrea et al., 2010).





in Alaska and found significantly fewer extreme minimum temperatures in the cold seasons and significantly more 5-day duration precipitation events. The impact of these changes on hydropower is that designers of new projects need to consider the possibility of heavier precipitation events and more surface runoff during fall, winter and early spring due to warmer nights. This could lead to the spillage of excess water without generation, if the reservoirs are capacity limited and already

full during those times of the year.

Extreme temperature and precipitation events may also link to sedimentation in reservoirs because, on principle, more force moves more sediment (Toniolo and Schultz, 2005). Other impacts of climate change on hydropower supply and demand include factors that affect runoff partitioning into surface and groundwater. These could be changes in the mean or extreme

events. These factors include climate-driven changes in vegetation and soil moisture. Studies of these trends, however, would depend on long-term, consistent, accurate, and co-located measurements of river discharge, vegetation, precipitation, and soil moisture, which are hard to find in the Far North.

## 2.3 Climate change complexity in the Far North

There are also complexities of changes in the Arctic, sub-Arctic, and alpine areas that are unique to the Far North. These include permafrost, sedimentation from thaw slumps, and glaciers. Ice rich permafrost acts as an aquatard, blocking passage of water through soil and limiting subsurface connectivity in groundwater (Carey et al., 2013).[3] As it thaws under a warming climate, subsurface storage and connectivity is anticipated to increase, but there is uncertainty about whether this will lead to more runoff and water availability for hydropower production and other hydroclimate impacts. For example, in areas of

currently continuous permafrost, permafrost could thaw disconnected areas and pull water down to the subsurface, but these could be in isolated voids and not part of a groundwater system that contributes to runoff. In areas of discontinuous permafrost, additional thawing is more likely to lead to changes in river discharge since the subsurface is more likely connected to runoff pathways. If surface waters infiltrate into the subsurface and soil is allowed to dry under either of these scenarios, there may be less moisture contributed to the regional atmosphere from the land surface, and less summer

precipitation, in turn.

Permafrost degradation in ice rich soil can also lead to thermakarst formation, in which thermal and water-driven erosion releases a significant volume of sediment into a water way, typically during a catastrophic collapse over a period of a few weeks to a couple of years. These types of sedimentation events can reduce reservoir capacity and increase wear and tear on

turbines in projects built adjacent to permafrost (Toniolo and Schultz, 2005; Gurnell, 1995). Newly built reservoirs will also impact permafrost distributions as the reservoirs fill and water has the potential to thaw additional soil in the newly formed

---

[3] Permafrost that is not ice rich may simply be frozen gravel or other material that still allows liquid to pass through to the deeper subsurface. Ice rich permafrost tends to be associated with big ice wedges and lenses, which are effective aquatards.





or enlarged lake.

Glacier dynamics are an additional complexity in northern climates. Glacial retreat is a widespread phenomenon throughout the Far North and many existing and proposed hydropower projects depend on river discharge from catchments with glaciers in the headwaters. As glaciers melt, they generate increasing discharge until this runoff reaches a peak, and declines thereafter (Fig. 6). Several researchers have suggested that most non-coastal glaciers are already past peak melt and that discharge from glacial sources is already declining (O'Neel et al., 2014, Radić and Hock, 2014; Arendt et al., 2002, 2009). Thus, the state and trajectory of glacial runoff is an important factor in designing hydropower projects and accurately estimating future reservoir inflow. The rates of melting glacial ice can also impact soil moisture and runoff partitioning: fast melt will lead to more saturated soil and surface runoff while slow melt is more likely to contribute to groundwater, so long as there is some discontinuity of the permafrost in the watershed. Finally, like permafrost degradation, retreating glaciers can leave behind large amounts of mobile sediment (Gurnell, 1995; Harrison et al., 1983), which can be transported into hydropower infrastructure and reduce the lifespans of turbines and reservoirs.

Climate change is also increasing the potential for glacial hazards. The melting of permafrost, along with glacial wastage, can lead to glacial lake outburst events (Bolsch et al., 2011), which have increased in number and magnitude in recent years (Sorg et al 2012; Narama et al., 2010; Horstmann, 2004; Agrawala 2003). These glacial hazards may be a threat to hydropower infrastructure (Richardson and Reynolds, 2000). In addition, glacial wastage is likely to increase the potential for flooding (Veijalainen et al., 2010; Kutuzov and Shahgedanova, 2009). While dams can be damaged by flooding, they can also be a way to mitigate flood risk by controlling water flow (World Bank, 2009).

Climate change is likely influencing the thickness of river and lake ice, as well as the timing of breakup, which has a significant impact on reservoir inflows. Numerous ice-related changes can be a challenge for hydropower operation, including a shift in patterns or frequency of ice blockage and jams (Gebre et al., 2013). Hydropower operators need to take into account these changing ice conditions, regardless of whether a project is run-of-river or has a reservoir (Prowse et al., 2011). Change in the duration and extent of ice cover is likely (Timalsina et al., 2013; Andrishak and Hicks, 2008). The later freeze up of lakes and rivers can lead to increased development of ice dam flooding or damage to infrastructure in the autumn (Molarius et al., 2010). Climate change has been shown to increase the frequency of spring ice jam floods in some regions (Bergstrom et al., 2001).

Polar climate is changing more rapidly than that of lower latitudes, a phenomenon known as Polar Amplification. The movements of pressure centers and the polar vortex, as well as changing sea ice cover, are impacting regional hydroclimatology. Researchers have described several mechanisms for Polar Amplification in the climate system. One of these mechanisms is the lowering of polar surface albedo as light-colored ice and snow melt, exposing dark ocean and land





surfaces, which in turn absorb more heat from solar radiation. Other reasons for Polar Amplification relate to the mass gradient in the atmosphere, from the equator to the poles, driven by atmospheric temperature and pressure. Heat and moisture tend to flow along this gradient, poleward. Projected rates of hydroclimate change in global- or low-latitude studies may underestimate the true rates of change for high latitudes in the future because of poor representation of these processes
in models (Clark et al., 2015; Christensen et al., 2013).

While we anticipate climate change will drive warmer, moister air poleward, air parcels are rarely transported meridionally. Instead, they are subject to Coriollis deflection and controlled by persistent lows and highs in the atmosphere. In the Far North, the Aleutian Low, Icelandic Low, Siberian High, and Polar Vortex are persistent patterns of circulation that help
direct atmospheric fronts and the movement of weather patterns that ultimately add up to climate. These persistent, covarying patterns of circulation, also known as ocean-atmosphere oscillations (such as the Pacific Decadal Oscillation, NAO, or ENSO) are considered modes of climate variability. While quantifying this variability with an oscillation index or another statistical tool can be useful for analyzing impacts, there are limits to how well the indices describe or predict the dynamics of persistent circulation patterns that actually determine local weather. In other words, while climate change theory
predicts a warmer, wetter north, that does not mean that more rain will necessarily arrive at a particular locality; it may be further to the East or West because of persistent circulation patterns.

Finally, other complexities in the Far North relate to social impacts of hydropower development in a changing climate. While the northern-most regions of the Earth are relatively sparsely populated, many indigenous and other rural people live
in communities that depend heavily on subsistence and commercial hunting and fishing, but also face high costs for electricity generated by non-renewable sources. Climate change is threatening the viability of food resources through physical, biological, and geochemical shifts. It is necessary to consider the additional impact that hydropower development and management of existing facilities has on the habitat of fish and game in a changing climate, even when it provides cheaper electricity.

### 3 Predicting Far Northern hydrology and estimating uncertainty

### 3.1 Long-term projections

Long-term climate projections from global climate models (GCMs) have now been recognized as valuable for both existing and future hydropower infrastructure planning and management (Viers, 2011). Traditional water supply projections lead to
different conclusions than projections that take into account climate change, including the range of plausible futures, or "uncertainty" around the model projections (Barsugli et al., 2012; Hamlet, 2011). The following general techniques are used in recent work to predict future hydrologic regimes (Clark et al., 2015; Hagemann et al., 2013; Barsugli et al., 2012; Chen et





al., 2012, 2014; Balsamo et al., 2009, 2011; Brekke et al., 2011; Koutsoyiannis et al., 2011; Lawrence and Hisdal, 2011; Frignon et al., 2007; Hagg et al., 2006; Bergstrom et al., 2001):

1.  Using a high quality baseline historical observational dataset that is at least 40 years long;

2.  Interpolating or "gridding" temperature and precipitation data to the spatial resolution of interest or assimilating these and other data into a reanalysis model to produce gridded fields

3.  Applying perturbations, various statistical techniques, or "error bars" around the observations, or using prior
10      knowledge to estimate the observational uncertainty;

4.  Running multiple GCMs, each either stand-alone or with a regional model nested in a global model, with a variety of different emissions scenarios and forcing data values (ensembles) for the historical period to represent observational uncertainty;

5.  Downscaling the GCM output to a local scale using statistical or dynamical methods;

6.  Comparing gridded baseline climatologic observations and model output for the historical period to evaluate model biases;

7.  Using the GCM output to force a hydrologic model or to perform a fully coupled simulation with river routing;

8.  Comparing hydrologic model output to runoff observations for the historical period.

25  These hydrological impact studies are the precursor to understanding impacts to hydropower (Jost and Weber, 2012; Pittock, 2010). Hydroclimate models can be used to understand glacier and snowmelt dynamics (Huss et al., 2008; Jonsdottir, 2008; Schaefli et al., 2007; Johannesson, 2006), but they can also be linked with energy market models to understand financial and technical feasibility; economic vulnerability to climate change (Cherry et al., 2005b; Harrison et al. 2003; Harrison and Whittington, 2002); and they can provide a methodological approach to understand tradeoffs (Rheinheimer et al., 2013).
30  Because getting to detailed, precise information at the watershed scale requires intensive effort and significant computing power, these methods have mostly been applied to case studies of particular basins (Bennett et al., 2012; Shrestha et al., 2012 in the Far North). Land surface models within GCMs and RCMs are another class of model that have been tested and compared for hydrologic applications, and are designed for global domains, but challenges of river routing (Li et al., 2013) and other local processes cause these tools to have large uncertainties at the watershed scale (Clark et al., 2015).



National assessments are important for understanding relative hydroclimate vulnerability (Rummukainen et al. 2003; Hurd et al., 1999; Lettenmaier et al. 1999), while basin-level models can provide more specific projections to inform water management (Döll et al., 2015; Frigon et al. 2007,). In several studies of the Colorado River Basin, it was found that under most projected climate scenarios, reduced flows in the river result in annual in-stream allocations being met less frequently, along with hydropower production declines (Vano et al., 2013; Rajagopalan et al., 2009; Christensen, 2004). Basin-level studies have also revealed shifts in flow timing and hydropower production (Finger et al., 2012). Hydropower modeling can provide specific information, at the plant level, to assess changing potential due to climate change (Chernet et al., 2013), and tailor management in order to maintain efficient production (Burn & Simonovic, 1996). Plant-level modeling can also integrate electricity market information in order to understand the economic impact of climate change on both the supply and demand for power (Gaudard et al., 2013).

### 3.2 Seasonal and shorter-term prediction

Northern states, countries, and individual utilities vary in the extent to which they use seasonal hydrologic prediction, which may reflect these organizations' existing capacity to utilize future projections as well (Inderberg and Løchen, 2012; Kirkinen et al., 2005). For example, the Alaska Pacific River Forecast Center (APRFC) will use NOAA's Climate Prediction Center seasonal outlook temperature and precipitation products as a general reference during the flood-prone spring river break-up season, but they do not use these products quantitatively to force hydrologic models during the season ahead. Hydro-Quebec and Norsk Hydro, large, semi-public utilities, on the other hand, have considerably more resources to use quantitative hydrologic modeling at the seasonal scale. In the US, climate or weather-model-driven seasonal hydrologic forecasting (SHF) is an emerging field, and one still limited to selected regions and a limited number of models (Yuan et al., 2015; Gochis et al., 2014). For hydropower management, however, quantitative seasonal forecasting should be routine and these model simulations will need to depend on a robust observational network. Ensemble approaches are a typical means for estimating uncertainty for seasonal forecasts and other researchers have innovated methods for quantifying uncertainty in the land surface models through Bayesian and other statistical techniques (Bevan et al., 2012; Clark et al., 2011).

Shorter-term, synoptic scale quantitative prediction of hydroclimate is of high value to facility managers in operations. However the accuracy of these forecasts is unknown in basins with few or no direct observations. With few observations and no quantitative forecasts, operators must rely on their past experiences and qualitative estimates of the impacts of synoptic weather events. During extreme events, such as an unprecedented rain-on-snow storm, managers may not have the past experience to anticipate the necessary operational decision-making. Short-term, coupled hydroclimate prediction systems could provide a valuable source of information for decision-makers. Like with longer-term projections, these too need uncertainty estimates.



### 3.3 Estimating and reducing uncertainty

For new, large hydropower projects, such as the proposed Susitna-Watana dam in Alaska, it is imperative to conduct a future hydroclimate projection study during the pre-licensing phase of the project to determine long-term water supplies and downstream impacts, given all of the climate change complexities noted above. Estimates of uncertainty, or the range of

plausible futures, either quantitative or qualitative, should be an essential component of these studies. Wolken et al. (2015) used a single climate model to assess glacial and hydrologic regimes, in a study that went far beyond what the Federal Energy Regulatory Commission (FERC) requested of the Alaska Energy Authority (AEA) during the Susitna-Watana pre-licensing phase. However the AEA did not fund a detailed uncertainty analysis component of the study. This leaves stakeholders with future runoff estimates but little sense of the uncertainty, or range, of these estimates. Brekke et al. (2011)

describes methods to include the uncertainty, or spread, projected by GCMs, the bias in climate models, and to account for local terrain and weather (for example through downscaling or high-resolution hydrologic modeling).

Two technological bottlenecks slow progress towards reducing uncertainty in this field: 1) accuracy and representativeness of hydroclimate observations in time and space and 2) quality and fidelity of models (Kundzewicz and Stakhiv, 2010;

Hawkins and Sutton, 2009;). These two issues are of particular concern in the Far North, because they also make it difficult to understand natural climate variability. Relative to the contiguous United States or mainland Europe, the Far North is particularly data poor, with respect to hydroclimate observations (Key et al., 2015; McClelland et al., 2015 and references therein). The shortcomings in estimates of ET and precipitation have already been discussed. Very little is known or understood about groundwater in the Far North, or how it might be changing (Bense et al., 2009). There are considerable

unknowns about the timing of glacial retreat, thawing of permafrost, and change in subsurface storage of water (Walsh et al., 2014; Murray et al, 2010).

Climate models also represent the climate system imperfectly, particularly the complexities in the Far North; this is another source of uncertainly. The global trajectory of greenhouse gas emissions into the future is another unknown. Many authors

have characterized and summarized these uncertainties over the past decade of literature (Eum et al. 2014; Kunreuther et al., 2013 and references therein; Lofgren et al. 2013; Clark et al., 2011).

In order to improve predictions of Far Northern hydrology and estimate uncertainty for hydropower planning we need to 1) improve, expand and sustain observational systems while 2) continuing to improve global and regional hydroclimate models

and output the necessary model parameters needed for decision-making. Observational systems include not just measurements of precipitation and discharge, but also topographic datasets and dynamic maps of subsurface features such as permafrost and groundwater. To improve models, process studies must occur in study basins, but then techniques learned in these efforts must propagate into the models. Local downscaling is an essential component of hydrologic projections, but 3)


a better quantitative evaluation of the uncertainty of these downscaled products is needed, based on emerging statistical techniques and ensemble simulations (Barsugli et al., 2013; Chen et al., 2012, 2014, Dibike et al., 2008).

## 4 Best practices for integration of climate change science into project planning and management

**4.1 Amass high quality information via baseline observations, process studies, and hydroclimate modeling**

For proposed and existing hydropower projects, project managers need to pull together a meaningful body of knowledge about the hydroclimate system, employing all of the techniques described in the preceding section. In doing so, it may be necessary to evaluate and potentially deploy additional observational systems and, if appropriate, keep them running over the lifespan of the project for decision support. At the current time, researchers are most confident about historical trends in

climate in the Far North that are derived from long-term, consistently-instrumented stations or remote sensing records (Curran et al., 2012). By and large, these observations are limited to air temperature, ground temperature, snow covered-area, and river discharge at just a few stations, or they have limited temporal coverage. From the subset of these records that cover at least 40 years or more, meaningful statistics can be generated about the historic patterns of variability and change. 60-80 years of data are even better. The more that observational networks expand and improve to accurately measure ET,

precipitation, groundwater, and water storage over long periods of time and in many locations, the more confident we will be in our knowledge of water resource availability.

Historic observations provide a basis to predict future change, given what we know about other mechanisms in the Earth System, which may affect how these future climate trajectories unfold. Unfortunately, the two years of river discharge

measurements required for a standard water use permit in the state of Alaska, for example, provide almost no information about the variability of that water source on interannual, decadal, and multi-decadal time scales. Models calibrated with long-term, high quality, historic data with fine temporal and spatial scales are the most robust, quantitative way to predict water resources into the future, as well as provide uncertainty estimates (Pechlivanidis et al., 2011).

Detailed hydrologic process studies in the project basin are also necessary in the Far North to ensure that appropriate models are developed and continue to be refined. These would include observations and modeling of small-scale systems impacting a resource watershed such as those from glacier, groundwater, permafrost, surface water, and meteorological inputs. These process studies help researchers and resource managers understand how watersheds respond to changes in temperature, precipitation, vegetation, etc. and give experts the confidence and information necessary to make qualitative and quantitative

predictions about future availability of water resources.

The modeling techniques described above in the long-term projections subsection should be employed to further push



regional hydroclimate modeling capabilities. While Earth system modeling has focused on regional model development, and polar modeling has made progress in this respect (e.g. Regional Arctic System Model, http://www.oc.nps.edu/NAME/RASM.htm), these fully coupled polar models are still relatively new and are used to simulate domains with a land surface resolution of ~50 km grid cells. Global Earth System models still do not simulate hydrology

realistically in the Far North, because they do not have detailed glacier, groundwater, and permafrost physics on the spatial scales necessary for water resource planning or management (Wolken et al., 2015). These physically based approaches are necessary to reduce uncertainty about hydrologic processes a decade or more into the future and can provide both qualitative and quantitative information. Decision makers need long-term hydroclimate projections over the managed basin updated at least every five years, as data records grow and the models advance. A release of each new generation of IPCC climate

model outputs, every 4-5 years, would be a logical trigger for new downscaled runoff estimates. This information is needed in the management of the water resources themselves, but also upstream and downstream impacts on habitat and ecosystems. For existing hydropower projects, hydroclimate projections should be used to consider structural, operational, and management adaptations, which are discussed in the next subsection.

## 4.2 Create structural and management adaptations for hydropower under climate change

A number of structural or operational adaptation practices could help integrate climate change science into project planning and management (ICOLD, 2013; Arsenault et al., 2013). These include:

1. *A more adaptive and regionalized hydropower facility licensing process.* The current licensing process in the United

States has not been responsive to climate change impacts (Viers, 2011), especially when management procedures such as storage and spill thresholds are firm and fixed. Several suggestions that have emerged in the literature are for adaptive management, shorter-term licenses, and more integrative licenses across basins. Explicit adaptive management of licenses could include specific operational responses to thresholds (Rheinheimer et al., 2013), more frequent assessment of performance, or adaptive licensing (ICOLD, 2013; Brekke et al., 2009; Madani, 2011). Brekke et al.

(2009) and these other studies reiterate that we can no longer assume a stationary hydrological future and licensing structures need to reflect this. Shorter contracts for hydropower licensing could allow operators to better consider and integrate climate change impacts on a regular basis (Viers, 2011). Reopening licenses is not an adequate adaptation; changes in hydrology that impact hydropower project operations can be projected at a useful level during project pre-licensing and re-licensing processes and should be addressed at those stages. Finally, it is important to promote

coordination of water management across projects so that cumulative effects within a basin are considered (Viers, 2011). This more regional approach would allow for greater balancing between ecological and production-related goals (Brown





et al., 2015; Viers, 2011, Marttila et al., 2005). Robust strategies in the licensing and operations may have an even bigger impact than making more precise predictions (Wilby, 2010).

2. *Flexible and adaptable operations rules.* One of the most consistent messages in the literature related to climate change and hydropower is that operations need to adapt their rules if they want to manage for optimal productivity (Gaudard et al., 2013; Vicuna et al., 2011; Madani 2010a, b; Minville, 2009, 2010; Raje and Mujumdar, 2010; Alfieri et al., 2006; Burn and Simonovic, 1996).  For example, operational rules should allow for management changes to facilities utilizing glacially fed basins, once the glaciers have significantly receded. If drought frequency increases thirty years after a project was built, a new operating paradigm is needed. This has been supported by studies that have shown that there is decreased performance of hydropower systems if management is rigid and does not adapt to changing conditions (Mehta et al., 2011; Perez-Diaz and Wilhelmi, 2010; Minville et al., 2009, 2010a,b; Schaefli et al., 2007; Robinson, 1997).

3. *Training for hydropower managers.* Water resource managers need support to more adequately incorporate climate change projections into the management of hydropower plants. Water managers often lack guidance about how to best integrate climate change into decision-making; detailed guidance and training should be developed through collaborations between FERC, hydropower trade organizations, and (internationally) the World Meteorological Organization, and the World Bank. Several studies have called for a better transfer of information from climate scientists to water managers (Lund, 2015; Milly et al., 2008; Oki, 2006;). This transfer can be accomplished by boundary organizations, which can bridge science and practice by translating projections in a way that can inform decision-making (Gordon et al., 2014; Miller et al., 2001, Cash et al., 2001, Lowrey et al., 2009;).  This could include higher resolution and more faithful models (i.e. more accurate for the correct physical reasons), a stable institutional platform for information exchange, and better communication about the precise needs of decision-makers. Training should also address better understanding about the uncertainty inherent in projections. Uncertainty is one of the reasons that climate forecasts are rarely used by managers (Rayner et al., 2005), despite these forecasts' utility. However, relying on the status quo of assuming hydrologic stationarity is counter-productive because the effects of climate change are evident already. While uncertainty cannot be fully overcome, it is possible to make wise management decisions despite this uncertainty (Ouranos, 2008).



### 4.3 Engage hydropower-supporting institutions to convey climate change guidance

There are several international organizations that are involved in different aspects of hydropower advocacy, information, development, and planning. These entities might be well-positioned to act as boundary organizations[4] to incorporate climate change information in project scoping and management. The International Hydropower Association (IHA) is a non-profit organization and global network working to advance sustainable hydropower. However, this organization has shown few tangible efforts to explore climate change impacts on hydropower. While IHA acknowledges that adaptation to climate change is a component of sustainable hydropower resource and reservoir planning (IHA, 2010), their recent conferences have had few sessions on climate change and hydropower. Those sessions that did address these topics focused more on climate change mitigation than climate change adaptation (IHA, 2013). Locher et al. (2010) reviewed IHA's recently published sustainability assessment protocol (IHA, 2010), and found that it pays insufficient attention to climate change and its impact on hydropower. Finally, the IHA (2013) report describes a new protocol for measuring carbon emissions from reservoirs, but has little other mention of climate change.

The World Commission on Dams (WCOD) is a global multi-stakeholder body initiated by the World Bank and World Conservation Union in response to opposition to large dam projects. A recent report discussed how climate change would likely reduce dam safety through an increase in extreme weather events (WCOD, 2000). They recommend that planning and ongoing monitoring should include modeling potential changes in flow due to climate change (WCOD, 2000). The International Centre for Hydropower (ICH) is an international association of companies and organizations that are active in all aspects of hydropower generation and supply. Recent ICH conference proceedings suggest that climate change will impact all hydropower development and cannot be ignored (ICH, 2016). Several sessions on hydropower and climate change suggest growing awareness of the effects of climate change on hydropower development and operations.

The International Energy Agency (IEA) is an organization that works to ensure reliable, clean and affordable energy for its 28 member countries. IEA tracks global energy statistics such as hydropower generation and consumption, making those data freely available for analysis. There has been a shift in their attention to climate change, but few concrete actions have been taken. In 2000, there was no clear direction or policy on climate change and hydropower within IEA according to their publications, while by 2013 it was mentioned as a key priority for moving forward within the organization (IEA 2013).

The World Bank is another organization that is involved in the funding of development projects around the globe, including hydropower. The World Bank has described how investments in the water sector are vulnerable to climate change (World

---

[4] These are organizations whose central purpose is to create and sustain meaningful and mutually beneficial links between knowledge producers and users (Meyer and Knight, 2014).





Bank, 2009). In response, they have stated that infrastructure should be designed using the best climate change information available, which should be taken into account when planning new projects (World Bank, 2009). They have found that traditional water supply projections lead to different conclusions than projections that take into account climate change (Ilimi 2007).

Finally, the International Commission on Large Dams (ICOLD) is a professional organization whose goal is to set guidelines and standards for building dams that are safe, efficient, economical, environmentally sound, and equitable. A technical committee is currently drafting a report on climate change, dams, reservoirs and water resources (ICOLD, 2013). This report, if completed, will describe the risks and uncertainties related to climate change and hydropower, present an impact

assessment framework, describe other drivers of change, discuss hydropower emissions, and provide suggestions and strategies for adaptation from the point of view of a pro-dam organization.

These types of hydropower-supporting organizations--regardless of how central advocacy is to their mission—could all benefit from having the best available information about climate change and helping convey it to stakeholders. Because

stakeholders are accustomed to getting information, guidance, and training from these organizations, extending that support to include climate change science could be highly effective.

## 5 Summary and conclusions

In summary:

1.  The climate of the Far North is changing. It is projected to get warmer and wetter here through the end of the century. Permafrost will continue to thaw, landlocked glaciers will continue to recede, and the surface/subsurface partitioning of water fluxes and storage will change. We have reviewed how some of these hydroclimate changes will likely impact hydropower in Far Northern regions, how this information relates to planning and management,

and we have described the state of the art techniques for predicting and estimating uncertainty.

2.  In order to improve predictions of Far Northern hydrology and estimating uncertainty for hydropower planning, it is necessary to improve, expand and sustain observational systems while continuing to improve global and regional hydroclimate models. A better quantitative evaluation of the uncertainty of these products is also needed, based on

emerging statistical techniques.

3.  Best practices in hydropower planning include having high quality information available to stakeholders from





observations, process studies, and hydroclimate modeling. It is imperative to create structural and management adaptations for hydropower under climate change. These include a more adaptive and regional hydropower facility licensing process, flexible and adaptable operations rules, and training for operators). Finally, hydropower-supporting institutions should be engaged as boundary organizations to convey climate change guidance.

4. We have shown that hydropower infrastructure is vulnerable to climate change. For new, large projects, such as the proposed Susitna-Watana dam in Alaska, it is essential to incorporate climate change, the complexities noted above, and uncertainty analysis into the research during the scoping phase to determine long-term water supplies and downstream impacts.

The changing climate and hydrology of the Far North is undoubtedly complex. While the hydoclimate research community is aware of shortcomings in models and datasets, over the past decade, techniques have emerged for hydrologic prediction and uncertainty analysis that are currently employable in water resource management. We need to continue to improve our baseline observations, do process studies that help improve models, quantify uncertainties where possible, and also identify when data is too sparse or of too poor quality to use for decision-making.

Best practices are just that: practices based on the best available information for hydropower planning and operations. Is this information perfect? Absolutely not, but the two pertinent questions are: 1) is this information worth the cost it takes to generate and 2) is it good enough for decision-making. For planned and existing projects worth millions or billions of dollars, hydroclimate prediction and uncertainty estimation require a tiny fraction of the cost of maintaining the facility. Given the potential impacts to hydropower production and efficiency, adaptation to climate change should be considered in the development of new hydropower projects (Kauna et al., 2012; Ouranos, 2008).

Is the best available climate change information good enough for decision-making? Yes. It is clear from the impacts of drought in the American West, and other regions, that climate change is already affecting existing hydropower projects (Konstantinos and Lettenmaier, 2006; Mote, 2006). For those facilities built 30 to 50 years ago, we did not have the technology to adequately project climate change. However, if hydropower managers were to ignore the best available hydroclimate information today, the cost to society could be quite high. Building flexible adaptations to climate change into hydropower planning and management is in the best interest of all who want to see these projects sustained or expanded.



## Author contribution

Cherry prepared the bulk of the manuscript and crafted the paper's outline and arguments. Knapp performed a literature review of many related social science and physical publications, which was woven throughout the paper. Trainor helped guide Knapp's literature review and helped polish the writing. Ray helped identify appropriate literature, refine the paper's arguments, and helped polish the writing. Tedesche performed a preliminary literature review focused on hydrology and helped proofread the final paper. Walker helped identify components to make the paper relevant for stakeholder decision-making and proofread the final paper.

## Acknowledgements

Cherry wishes to thank the National Oceanic and Atmospheric Administration-National Marine Fisheries Service (NOAA-NMFS) for financial support of this effort (Award # HA-133F-12-SE-2460) and Lily Cohen at the University of Alaska Fairbanks (UAF) for helping organize the references. Tedesche, Knapp, and Trainor acknowledge financial support from NOAA-NMFS under that same award number. Trainor also acknowledges support from NOAA Climate Program Office Grant NA11OAR4310141 through the Alaska Center for Climate Assessment and Policy at UAF, Alaska EPSCoR NSF award #OIA-1208927, and the state of Alaska. Ray's participation in this effort was supported in-kind by the NOAA/ESRL Physical Sciences Division and Walker was supported in-kind by NOAA-NMFS.

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





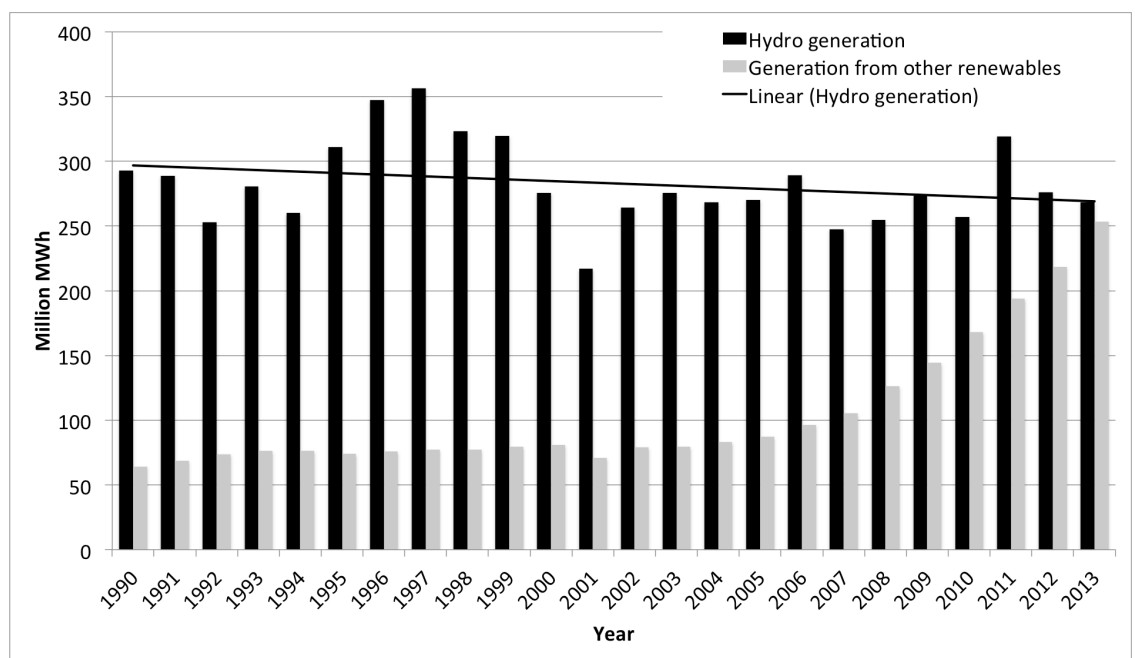

**Figure 1: Trends in hydropower production and other renewables in the United States. Data are from http://www.eia.gov/electricity/annual/**

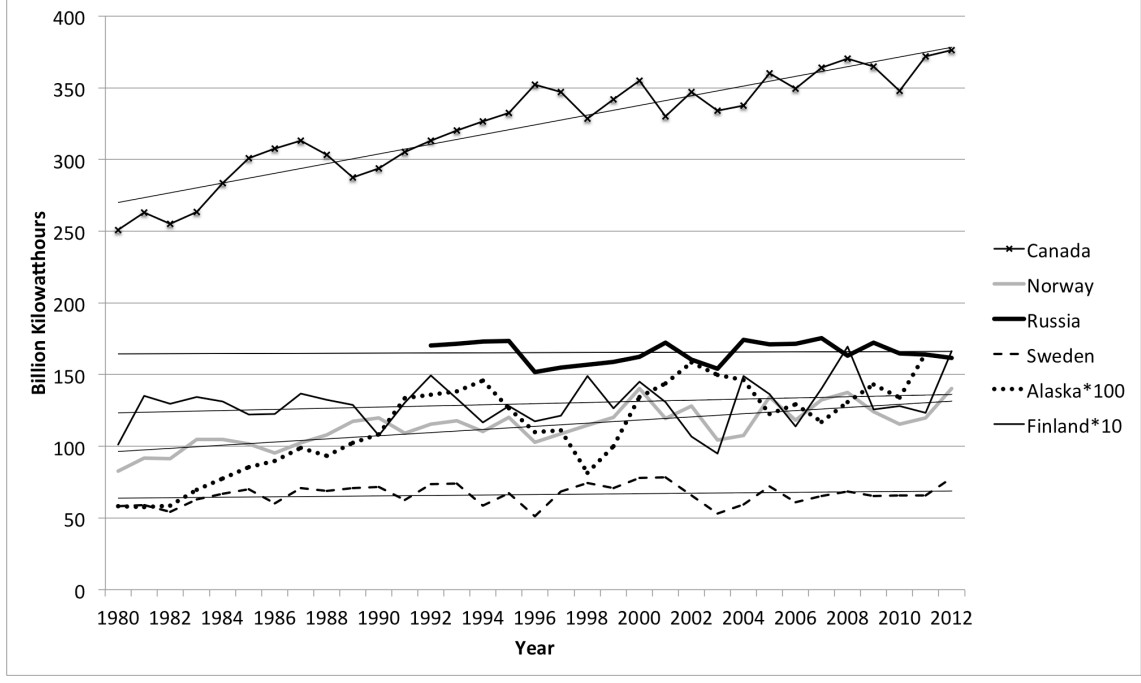

**Figure 2: Trends in hydropower generation in selected northern regions, 1980-2012. Data are from the Energy Information Administration and the Alaska Energy Data Gateway.**





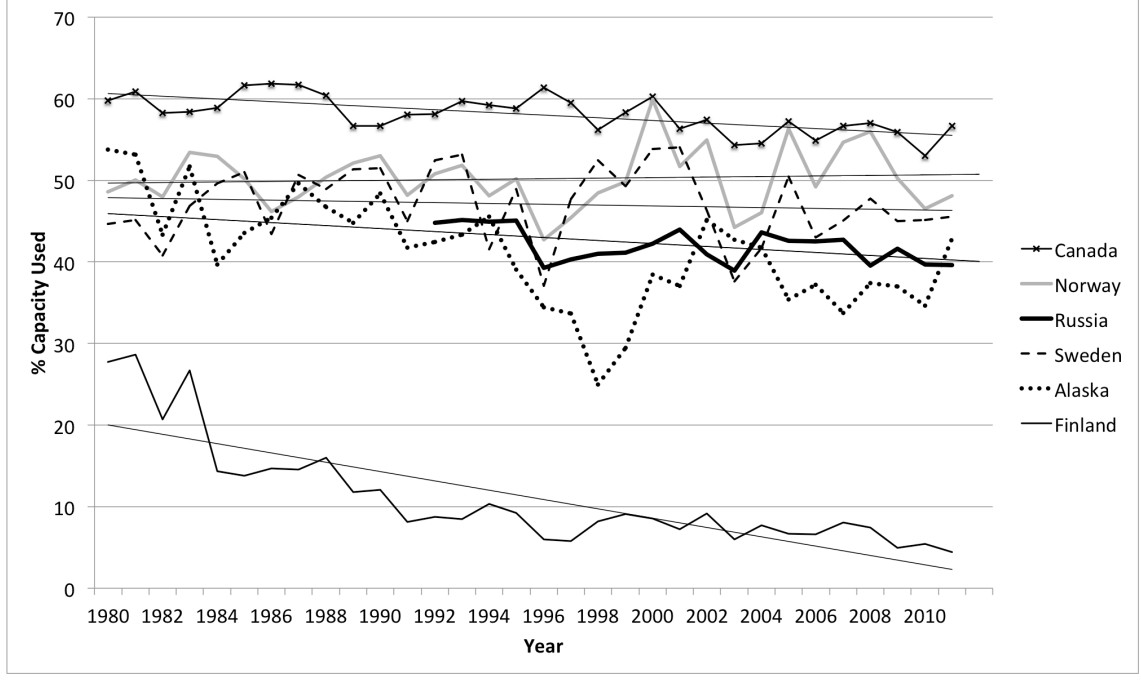

**Figure 3: Trends in hydropower capacity in selected northern regions, 1980-2012. Data are from the Energy Information Administration and the Alaska Energy Data Gateway.**

**Figure 4: Percent capacity of hydropower used in selected northern regions, 1980-2012. Data are from the Energy Information Administration and the Alaska Energy Data Gateway.**




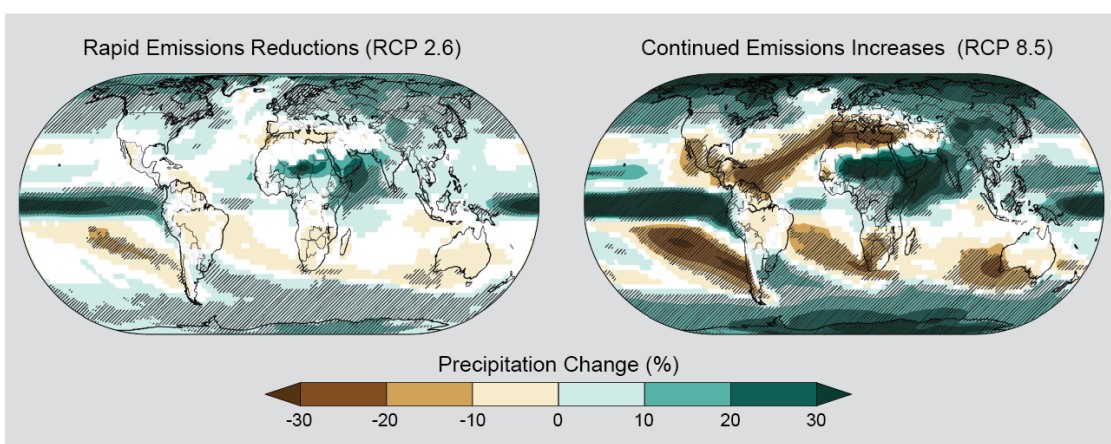

**Figure 5: Projected change in average Annual Precipitation from the 3[rd] National Climate Assessment (Walsh et al., 2014).**

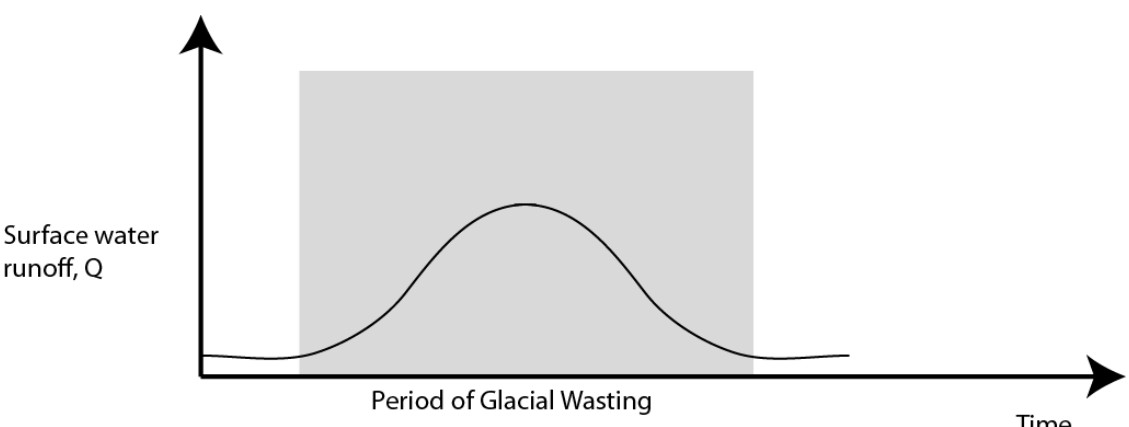

**Figure 6: Conceptual drawing of surface water runoff as a function of time when glaciers are wasting. Modified from Jansson et al. 2003.**