# Peer review of "Planning for climate change impacts on hydropower in the Far North"

_Hydrology and Earth System Sciences, 2016_

## Referee Comment (RC1) · Anonymous Referee #1 · 1 Jul 2016

This article engages an important topic–integrating climate change impacts on hydrologic systems into hydropower planning and management in the Far North regions of the world. The article is a well-written synthesis of current literature and an adequate presentation of best practices and recommendations for further incorporating climate data, predictions, and uncertainty analysis into hydropower planning, licensing, and operations management.

I have three major concerns about the manuscript and suggestions for revision that are likely easily addressed:

1) There are no specific methods indicated that convey any rigor or robustness in your literature review. The contribution of this article is very much tied to your literature review and the synthesis thereof. Thus, I would expect (likely in Section 2.1, or perhaps

in a standalone methods section) that the authors describe "how" this literature review was approach and what methods were employed to ensure that the literature review was comprehensive, robust, and replicable. This section does not have to be long, likely a paragraph, but it is necessary to comment, for example, on what databases were employed; what key words, phrases, or text combinations were searched; how the literature was organized; and if/how the detailed analysis of literature was approached, e.g., deductively, inductively, or both. If both, what important themes were emergent from the literature as opposed to those already conceptualized by the authors. A discussion of this caliber is necessary to ensure the reader that your review is comprehensive, and thus your findings more meaningful.

2) There is little to no context provided for the proposed Susitna-Watana dam project that you reference in the article. As far as I can tell, you reference this proposed project briefly, only three times in the text: lines 18-19 on page 2; lines 2-9 on page 11; and in the conclusion, page 17, line 7. You also cite it as a keyword–which I think at this point is misleading. It seems to me that your reference to the project is an attempt to ground your findings and suggestions in a real-time need and potential application. If that is the case, great, but for the reader's sake (who, given your larger scope of the "Far North," will likely not just be from Alaska) please include appropriate context describing the project, including its specific geographic location; important biophysical parameters (river flow, potential storage, potential generation capacity); current project status (planning phase, licensing, construction, etc.); proposed biophysical and social impacts; and even perhaps a map. Currently, there just is not enough information provided for the reference to this proposed project to be meaningful, and instead, it is a distraction. One option is to remove reference to the project altogether. Another would be to keep it as an example, specifically on page 11, but to use it as an opportunity to expand on the current process for developing hydropower projects (specifically in the U.S.) and how/when further estimating and reducing uncertainty could be better built into the process. This approach could potentially hold true for the proposed "best practices" as well. The Susitna-Watana project context could be used to describe

"how" the proposed best practices might be operationalized in the planning stages of hydropower development in the Far North. The way the authors currently reference the project on page 2 and in the keywords may lead the reader to believe that this is a case study that will be explored within the article, and this is not the case.

3) Best practices could (and in my opinion should) be further discussed in terms of "how" they may be operationalized. Your reference to engaging boundary organizations for this purpose is not adequate. For example, in Section 4.1, who do you propose would organize and pay for this type and level of instrumentation, monitoring, and ongoing analysis? Would this be a term of licensing imposed on the operator? What about in Far North countries (e.g., Russia) where there may be a lack of political will to impose such terms? Would this instead be undertaken by governments in preparation for licensing? Is there political will and enough economic incentive to support this? (I realize your argument about the long-term economic viability of these best practices, but funding decisions for this type of work are made by political actors who mostly rely on short-term economic gains to retain public office.) While I agree that utilizing boundary organizations spanning stakeholders (from operators to regulators) is a novel approach to bring data monitoring and analysis capacity (and transparency) to this complex problem in a more flexible and adaptive manner, your approach to conveying this in the article did not comment on "how" this might occur. You simply referenced the existing organizations that could potentially serve as boundary organizations. How should stakeholders at different scales best engage these resources? While the answer is obviously context dependent, are there good examples in the literature of how one or more of these organizations is current serving in this capacity in the Far North? Could you employ an example to support your suggestion and the underlying assumption that this would work to support the integration of climate change information into hydropower planning?

Minor comments aimed at improving the manuscript for publication:

-Page 2, lines 16-18: does Figure 4 only suggest "the impacts of a changing climate,

or a misalignment of infrastructure and resources, or both"? Could Figure 4 not also suggest some of the issues you raised earlier in the introduction such as in-stream flow regulations (page 1, line 31) as well as the associated social concern about impacts to related biophysical resources such as aquatic habitat and ecosystem diversity and connectivity, which includes the political to protect these qualities in certain geographies.

-Page 3, line 4: have you established that there are "barriers to th[e] process" of "integrat[ing] climate change science into hydropower management"? Can you make these barriers more explicit, either here in the introduction or elsewhere in the paper?

-Page 3, lines 20-26: I suggest either condensing this paragraph and adding it to the introduction or removing altogether as unnecessary.

-Section 2.2: I think this section could be substantially shortened. Condense your review of these studies into important, concise points and relate them more directly to hydropower as you do in Section 2.3 (see page 6, lines 29-30 for what I think is a great example of how you explicitly tied the literature review of climate impacts directly to the hydropower discussion that is the focus of this article).

-Page 4, line 23: remove all unnecessary, colloquial text such as "Suffice it to say. . .." See also, "By and large" (page 12, line 11), and the rhetorical question and answers of page 17, lines 17 and 24.

-Page 4, line 23: what is the purpose of the phrase "from first principles" here? This is unclear.

-Page 7, line 5: perhaps insert "yearly" after "increasing" to add clarity to this concept.

-Page 7, lines 31-34: these concepts require citations within the paragraph, not just at the end.

-Page 9, lines 4-23: I encourage you to align citations (from lines 1-2) with the specific techniques (or multiple) that they are most associated with. This will help readers more

easily identify specific citations of interest for further investigation with regard to these specific tools/techniques.

-Page 10, line 16: is this the first time the acronym NOAA is used? If so, please spell out.

-Page 11, lines 2-9: as previously mentioned, more explanation of the U.S. hydropower licensing process here might be helpful for international readers–if you decide to keep this example.

-Page 13, line 24: how do you define "adaptive licensing" in this context? How is it different from adaptive management built into (or as a condition of) the licensing process/actual license?

-Page 14, line 1: what do you mean by "Robust strategies" in this sentence? This seems vague.

-Page 14, line 20: why wouldn't you place footnote #4 here, upon first mention of the term "boundary organizations"?

-Figures 2-3: what is the purpose of "*100" and "*10" behind "Alaska" and "Finland" respectively in the legends of these figures?

-Figure 5: what does the grey-striated (diagonally-stripped) area of this map mean? It is unclear from the legend and Figure caption.

-Figure 6: could you replace "drawing" in the figure caption with "graph" for clarity?

[Figure]

---

## Referee Comment (RC2) · Anonymous Referee #2 · 30 Jul 2016

Comment 1: It is imperative that planners and decision-makers have access to information on uncertainty in not only flows but also in sediment loads so these can be accounted for in the design of new hydropower projects. Change in climate can influence erosion and catchment sediment yield. This change in sediment yield can affect the reservoir storage volume and power production. If this is relevant to the Far North then, I think author should develop some paragraph on this aspect too.

Comment 2: In general there are three major sources of uncertainty related to climate change predictions: (1) due to GCM, emission scenarios and the down-scaling method used (2) uncertainty in land use change which is mostly neglected but should be included and (3) uncertainty due to hydrological modeling. These sources of uncertainties make hydrological predictions challenging. Nevertheless quantifying the uncertainty at every step in the modelling process (cascading uncertainty) can address

the challenge in quantitative assessment of climate change impacts on catchment hydrology considering the full range of uncertainties involved. Hence what is needed is a careful investigation of sources of uncertainty through cascading and then identification of the major source(s) of uncertainty and focus on reducing that major source(s) of uncertainty. There are recent studies which have pointed out that major source of uncertainty can vary with time and is space dependent. Some discussion on these points could strengthen more Section 3.3 Estimating and reducing uncertainty.

---

## Author Comment (AC1) · 20 Aug 2016

To: Editorial Board and Discussion Forum, HESS Manuscript: Planning for climate change impacts on hydropower in the Far North Number: doi:10.5194/hess-2016-167

From: Jessica Cherry Research Associate Professor University of Alaska Fairbanks And Co-authors Corrie Knapp, Sarah Trainor, Andrea Ray, Molly Tedesche, and Susan Walker

August 19, 2016

Reviewer One's comments:

REVIEWER: This article engages an important topic–integrating climate change impacts on hydro- logic systems into hydropower planning and management in the Far

North regions of the world. The article is a well-written synthesis of current literature and an adequate presentation of best practices and recommendations for further incorporating climate data, predictions, and uncertainty analysis into hydropower planning, licensing, and operations management. I have three major concerns about the manuscript and suggestions for revision that are likely easily addressed: 1) There are no specific methods indicated that convey any rigor or robustness in your literature review. The contribution of this article is very much tied to your literature re- view and the synthesis thereof. Thus, I would expect (likely in Section 2.1, or perhaps in a standalone methods section) that the authors describe "how" this literature review was approach and what methods were employed to ensure that the literature review was comprehensive, robust, and replicable. This section does not have to be long, likely a paragraph, but it is necessary to comment, for example, on what databases were employed; what key words, phrases, or text combinations were searched; how the literature was organized; and if/how the detailed analysis of literature was approached, e.g., deductively, inductively, or both. If both, what important themes were emergent from the literature as opposed to those already conceptualized by the authors. A discussion of this caliber is necessary to ensure the reader that your review is comprehensive, and thus your findings more meaningful.

AUTHORS: The authors would be happy to add a paragraph describing the methods behind the literature search. Here's a proposed paragraph addition, inspired by the reviewer's comment: Literature was analyzed using an inductive approach; an incomplete set of prior hydrologic studies have been performed in any region, let alone, the Far North, but specific examples were used to construct more generalized best practices. Google's default search engine, as well as Google Scholar, and ISI's Web of Science were scanned with terms: hydrology, hydrologic modeling, water resources, glaciers, groundwater, water and climate change, hydropower, hydroelectric power, with and without regional qualifiers such as Canada, Russia, Scandinavia (and specific countries therein), and Alaska. Large publications in the field such as those by the Intergovernmental Panel on Climate Change author teams and the National Climate
Assessment author teams were searched for relevant material. Resulting articles from peer-reviewed and gray literature were cataloged in a database. Analysis focused on peer-reviewed and government-authored reports, and viewed trade publications (for example from the hydropower industry) as potentially biased.

REVIEWER: 2) There is little to no context provided for the proposed Susitna-Watana dam project that you reference in the article. As far as I can tell, you reference this proposed project briefly, only three times in the text: lines 18-19 on page 2; lines 2-9 on page 11; and in the conclusion, page 17, line 7. You also cite it as a keyword–which I think at this point is misleading. It seems to me that your reference to the project is an attempt to ground your findings and suggestions in a real-time need and potential application. If that is the case, great, but for the reader's sake (who, given your larger scope of the "Far North," will likely not just be from Alaska) please include appropriate context describing the project, including its specific geographic location; important biophysical parameters (river flow, potential storage, potential generation capacity); current project status (planning phase, licensing, construction, etc.); proposed biophysical and social impacts; and even perhaps a map. Currently, there just is not enough information provided for the reference to this proposed project to be meaningful, and instead, it is a distraction. One option is to remove reference to the project altogether. Another would be to keep it as an example, specifically on page 11, but to use it as an opportunity to expand on the current process for developing hydropower projects (specifically in the U.S.) and how/when further estimating and reducing uncertainty could be better built into the process. This approach could potentially hold true for the proposed "best practices" as well. The Susitna-Watana project context could be used to describe "how" the proposed best practices might be operationalized in the planning stages of hydropower development in the Far North. The way the authors currently reference the project on page 2 and in the keywords may lead the reader to believe that this is a case study that will be explored within the article, and this is not the case.

AUTHORS: The authors wish to respond by adding more information about the pro-

posed Susitna-Watana dam project, as the reviewer suggested. In particular, the project location, details on size, physical parameters, project status would be added, as well as a short description of the licensing process in the US and at which stage the proposed best practices might be implemented.

REVIEWER: 3) Best practices could (and in my opinion should) be further discussed in terms of "how" they may be operationalized. Your reference to engaging boundary organizations for this purpose is not adequate. For example, in Section 4.1, who do you propose would organize and pay for this type and level of instrumentation, monitoring, and ongoing analysis? Would this be a term of licensing imposed on the operator? What about in Far North countries (e.g., Russia) where there may be a lack of political will to impose such terms? Would this instead be undertaken by governments in preparation for licensing? Is there political will and enough economic incentive to support this? (I realize your argument about the long-term economic viability of these best practices, but funding decisions for this type of work are made by political actors who mostly rely on short-term economic gains to retain public office.) While I agree that utilizing boundary organizations spanning stakeholders (from operators to regulators) is a novel approach to bring data monitoring and analysis capacity (and transparency) to this complex problem in a more flexible and adaptive manner, your approach to conveying this in the article did not comment on "how" this might occur. You simply referenced the existing organizations that could potentially serve as boundary organizations. How should stakeholders at different scales best engage these resources? While the answer is obviously context dependent, are there good examples in the literature of how one or more of these organizations is current serving in this capacity in the Far North? Could you employ an example to support your suggestion and the underlying assumption that this would work to support the integration of climate change information into hydropower planning?

AUTHORS: The authors think that a strategy of who should pay for these studies is beyond the scope of this paper (we aim to be decision supporters not decision makers)

and a one-fit solution is unlikely in all Far Northern regions. We can however discuss this further in section 4.1 including the implications, for example, of having the federal government versus the utility or another stakeholder cover the cost of ongoing monitoring. We can also provide examples of how boundary organizations are currently providing information services in the water resource sector.

REVIEWER: Minor comments aimed at improving the manuscript for publication: - Page 2, lines 16-18: does Figure 4 only suggest "the impacts of a changing climate, or a misalignment of infrastructure and resources, or both"? Could Figure 4 not also suggest some of the issues you raised earlier in the introduction such as in-stream flow regulations (page 1, line 31) as well as the associated social concern about impacts to related biophysical resources such as aquatic habitat and ecosystem diversity and connectivity, which includes the political to protect these qualities in certain geographies.

AUTHORS: Yes, this is true and we can add this clarification.

REVIEWER:-Page 3, line 4: have you established that there are "barriers to th[e] process" of "integrat[ing] climate change science into hydropower management"? Can you make these barriers more explicit, either here in the introduction or elsewhere in the paper?

AUTHORS: Yes, we can add clarification here.

REVIEWER:-Page 3, lines 20-26: I suggest either condensing this paragraph and adding it to the introduction or removing altogether as unnecessary.

AUTHORS: We plan to condense this paragraph.

REVIEWER:-Section 2.2: I think this section could be substantially shortened. Condense your review of these studies into important, concise points and relate them more directly to hydropower as you do in Section 2.3 (see page 6, lines 29-30 for what I think is a great example of how you explicitly tied the literature review of climate impacts

directly to the hydropower discussion that is the focus of this article).

AUTHORS: Yes, we can condense this section.

REVIEWER:-Page 4, line 23: remove all unnecessary, colloquial text such as "Suffice it to say. . .." See also, "By and large" (page 12, line 11), and the rhetorical question and answers of page 17, lines 17 and 24.

AUTHORS: We can do this.

REVIEWER:-Page 4, line 23: what is the purpose of the phrase "from first principles" here? This is unclear.

AUTHORS: We meant from basic thermodynamics and atmospheric dynamics on a rotating planet. We will remove this or clarify it.

REVIEWER:-Page 7, line 5: perhaps insert "yearly" after "increasing" to add clarity to this concept.

AUTHORS: We can do this.

REVIEWER:-Page 7, lines 31-34: these concepts require citations within the paragraph, not just at the end.

AUTHORS: We can do this.

REVIEWER:-Page 9, lines 4-23: I encourage you to align citations (from lines 1-2) with the specific techniques (or multiple) that they are most associated with. This will help readers more easily identify specific citations of interest for further investigation with regard to these specific tools/techniques.

AUTHORS: We can do this.

REVIEWER:-Page 10, line 16: is this the first time the acronym NOAA is used? If so, please spell out.

AUTHORS: We can do this.

REVIEWER:-Page 11, lines 2-9: as previously mentioned, more explanation of the U.S. hydropower licensing process here might be helpful for international readers–if you decide to keep this example.

AUTHORS: We can do this.

REVIEWER:-Page 13, line 24: how do you define "adaptive licensing" in this context? How is it different from adaptive management built into (or as a condition of) the licensing process/actual license?

AUTHORS: We use these terms interchangeably, but can clarify this in the text.

REVIEWER:-Page 14, line 1: what do you mean by "Robust strategies" in this sentence? This seems vague.

AUTHORS: We can clarify this. We mean a strategy that will last for more than a year or two under a variety of different climate and economic conditions, i.e. adaptive management.

REVIEWER:-Page 14, line 20: why wouldn't you place footnote #4 here, upon first mention of the term "boundary organizations"?

AUTHORS: We can change that.

REVIEWER:-Figures 2-3: what is the purpose of "*100" and "*10" behind "Alaska" and "Finland" respectively in the legends of these figures?

AUTHORS: The units have been multiplied by these factors to fit them all on the same graph. We can change this to make is more clear.

REVIEWER:-Figure 5: what does the grey-striated (diagonally-stripped) area of this map mean? It is unclear from the legend and Figure caption.

AUTHORS: We will add: Hatched areas indicate statistically consistent results and consistency across models.
**HESSD**

REVIEWER:-Figure 6: could you replace "drawing" in the figure caption with "graph" for clarity?

AUTHORS: Yes, we can do that.

Reviewer Two's comments:

REVIEWER: Comment 1: It is imperative that planners and decision-makers have access to information on uncertainty in not only flows but also in sediment loads so these can be accounted for in the design of new hydropower projects. Change in climate can influence erosion and catchment sediment yield. This change in sediment yield can affect the reservoir storage volume and power production. If this is relevant to the Far North then, I think author should develop some paragraph on this aspect too.

AUTHORS: Agreed: we can add a short statement on the value of climate change-driven sediment load modeling.

REVIEWER: Comment 2: In general there are three major sources of uncertainty related to climate change predictions: (1) due to GCM, emission scenarios and the downscaling method used (2) uncertainty in land use change which is mostly neglected but should be included and (3) uncertainty due to hydrological modeling. These sources of uncertainties make hydrological predictions challenging. Nevertheless quantifying the uncertainty at every step in the modelling process (cascading uncertainty) can address the challenge in quantitative assessment of climate change impacts on catchment hydrology considering the full range of uncertainties involved. Hence what is needed is a careful investigation of sources of uncertainty through cascading and then identification of the major source(s) of uncertainty and focus on reducing that major source(s) of uncertainty. There are recent studies which have pointed out that major source of uncertainty can vary with time and is space dependent. Some discussion on these points could strengthen more Section 3.3 Estimating and reducing uncertainty.

AUTHORS: Many individual authors and the IPCC teams have categorized climaterelated uncertainty in different ways and we've tried to describe some of these. We can add cascading uncertainty as another issue. There are some uncertainties that are specific to the Far North (changes in land use are typically not as large as in mid-latitudes, but sparse observations are a bigger problem), but we can add a discussion of the reviewer's point to this section 3.3.

---

## Author Response (AR1)

To: Editorial Board, HESS
Manuscript: Planning for climate change impacts on hydropower in the Far North
Number: doi:10.5194/hess-2016-167

From: Jessica Cherry
Research Associate Professor
University of Alaska Fairbanks
And Co-authors Corrie Knapp, Sarah Trainor, Andrea Ray, Molly Tedesche, and Susan Walker

November 2, 2016

Dear Editorial Board,

Thanks for giving us the opportunity to make these revisions to our paper. We think the paper is better because of them, of course.

Please don't hesitate to contact us if you have further questions.

Thanks,
Jessica Cherry, on behalf of all of the authors

**Reviewer One's comments:**

REVIEWER: This article engages an important topic–integrating climate change impacts on hydrologic systems into hydropower planning and management in the Far North regions of the world. The article is a well-written synthesis of current literature and an adequate presentation of best practices and recommendations for further incorporating climate data, predictions, and uncertainty analysis into hydropower planning, licensing, and operations management.

I have three major concerns about the manuscript and suggestions for revision that are likely easily addressed:

1) There are no specific methods indicated that convey any rigor or robustness in your literature review. The contribution of this article is very much tied to your literature review and the synthesis thereof. Thus, I would expect (likely in Section 2.1, or perhaps in a standalone methods section) that the authors describe "how" this literature review was approach and what methods were employed to ensure that the literature review was comprehensive, robust, and replicable. This section does not have to be long, likely a paragraph, but it is necessary to comment, for example, on what databases were employed; what key words, phrases, or text combinations were searched; how the literature was organized; and if/how the detailed analysis of literature was approached, e.g., deductively, inductively, or both. If both, what important themes were emergent from the literature as opposed to those already conceptualized by the authors. A discussion of this caliber is necessary to ensure the reader that your review is comprehensive, and thus your findings more meaningful.

"Literature was analyzed using an inductive approach; an incomplete set of prior hydrologic studies have been performed in any region, let alone, the Far North, but specific examples were used to construct more generalized best practices. Google's default search engine, as well as Google Scholar, and ISI's Web of Science were scanned with terms: hydrology, hydrologic modeling, water resources, glaciers, groundwater, water and climate change, hydropower, hydroelectric power, with and without regional qualifiers such as Canada, Russia, Scandinavia (and specific countries therein), and Alaska. Large publications in the field such as those by the Intergovernmental Panel on Climate Change author teams and the National Climate Assessment author teams were searched for relevant material. Resulting articles from peer-reviewed and gray literature were cataloged in a database. Analysis focused on peer-reviewed and government-authored reports, and viewed trade publications (for example from the hydropower industry) as potentially biased. "

The above paragraph was added to section 2.1 now called '**2.1 Scope and Methods of Study**'.

REVIEWER: 2) There is little to no context provided for the proposed Susitna-Watana dam project that you reference in the article. As far as I can tell, you reference this proposed project briefly, only three times in the text: lines 18-19 on page 2; lines 2-9 on page 11; and in the conclusion, page 17, line 7. You also cite it as a keyword–which I think at this point is misleading. It seems to me that your reference to the project is an attempt to ground your findings and suggestions in a real-time need and potential application. If that is the case, great, but for the reader's sake (who, given your larger scope of the "Far North," will likely not just be from Alaska) please include appropriate context describing the project, including its specific geographic location; important biophysical parameters (river flow, potential storage, potential generation capacity); current project status (planning phase, licensing, construction, etc.); proposed biophysical and social impacts; and even perhaps a map. Currently, there just is not enough information provided for the reference to this proposed project to be meaningful, and instead, it is a distraction. One option is to remove reference to the project altogether. Another would be to keep it as an example, specifically on page 11, but to use it as an opportunity to expand on the current process for developing hydropower projects (specifically in the U.S.) and how/when further estimating and reducing uncertainty could be better built into the process. This approach could potentially hold true for the proposed "best practices" as well. The Susitna-Watana project context could be used to describe "how" the proposed best practices might be operationalized in the planning stages of hydropower development in the Far North. The way the authors currently reference the project on page 2 and in the keywords may lead the reader to believe that this is a case study that will be explored within the article, and this is not the case.

The key words were removed and but the authors did add more information on the Susitna-Watana Dam throughout the text (particularly on page 11).

REVIEWER: 3) Best practices could (and in my opinion should) be further discussed in

terms of "how" they may be operationalized. Your reference to engaging boundary organizations for this purpose is not adequate. For example, in Section 4.1, who do you propose would organize and pay for this type and level of instrumentation, monitoring, and ongoing analysis? Would this be a term of licensing imposed on the operator? What about in Far North countries (e.g., Russia) where there may be a lack of political will to impose such terms? Would this instead be undertaken by governments in preparation for licensing? Is there political will and enough economic incentive to support this? (I realize your argument about the long-term economic viability of these best practices, but funding decisions for this type of work are made by political actors who mostly rely on short-term economic gains to retain public office.) While I agree that utilizing boundary organizations spanning stakeholders (from operators to regulators) is a novel approach to bring data monitoring and analysis capacity (and transparency) to this complex problem in a more flexible and adaptive manner, your approach to conveying this in the article did not comment on "how" this might occur. You simply referenced the existing organizations that could potentially serve as boundary organizations. How should stakeholders at different scales best engage these resources? While the answer is obviously context dependent, are there good examples in the literature of how one or more of these organizations is current serving in this capacity in the Far North? Could you employ an example to support your suggestion and the underlying assumption that this would work to support the integration of climate change information into hydropower planning?

The authors think that a strategy of who should pay for these studies is beyond the scope of this paper (we aim to be decision supporters not decision makers) and a one-fit solution is unlikely in all Far Northern regions. We did however discuss this further in section 4.2.

REVIEWER: Minor comments aimed at improving the manuscript for publication:

-Page 2, lines 16-18: does Figure 4 only suggest "the impacts of a changing climate, or a misalignment of infrastructure and resources, or both"? Could Figure 4 not also suggest some of the issues you raised earlier in the introduction such as in-stream flow regulations (page 1, line 31) as well as the associated social concern about impacts to related biophysical resources such as aquatic habitat and ecosystem diversity and connectivity, which includes the political to protect these qualities in certain geographies.

We added this clarification.

REVIEWER:-Page 3, line 4: have you established that there are "barriers to th[e] process" of "integrat[ing] climate change science into hydropower management"? Can you make these barriers more explicit, either here in the introduction or elsewhere in the paper?

We added more detail on resistance to doing this by the US Federal Energy Regulatory Commission later in the paper.

REVIEWER:-Page 3, lines 20-26: I suggest either condensing this paragraph and adding it to the introduction or removing altogether as unnecessary.

A citation has been added to this paragraph, but a couple of our water resource managers provided feedback that this was an important paragraph to leave in.

REVIEWER:-Section 2.2: I think this section could be substantially shortened. Condense your review of these studies into important, concise points and relate them more directly to hydropower as you do in Section 2.3 (see page 6, lines 29-30 for what I think is a great example of how you explicitly tied the literature review of climate impacts directly to the hydropower discussion that is the focus of this article).

The section is really one of the hearts of the paper and we've decided not to shorten it substantially, though a couple of sentences were removed or shortened. Readers who are unfamiliar with cold regions will not necessarily be familiar with these issues.

REVIEWER:-Page 4, line 23: remove all unnecessary, colloquial text such as "Suffice it to say. . .." See also, "By and large" (page 12, line 11), and the rhetorical question and answers of page 17, lines 17 and 24.

We have removed the colloquial language but believe that the summary and conclusions are an appropriate place for the questions asked and answered in this section.

REVIEWER:-Page 4, line 23: what is the purpose of the phrase "from first principles" here? This is unclear.

We have clarified what we mean from 'first principles'. This is a common term in climate physics.

REVIEWER:-Page 7, line 5: perhaps insert "yearly" after "increasing" to add clarity to this concept.

We added the word 'annual' to clarify this statement.

REVIEWER:-Page 7, lines 31-34: these concepts require citations within the paragraph, not just at the end.

We've added a reference for a textbook on climate; the process is described in most textbook on climate physics.

REVIEWER:-Page 9, lines 4-23: I encourage you to align citations (from lines 1-2) with the specific techniques (or multiple) that they are most associated with. This will help readers more easily identify specific citations of interest for further investigation with regard to these specific tools/techniques.

We tried this but it cluttered the section and was highly repetitive, so we took it back out.

REVIEWER:-Page 10, line 16: is this the first time the acronym NOAA is used? If so, please spell out.

This was done.

REVIEWER:-Page 11, lines 2-9: as previously mentioned, more explanation of the U.S. hydropower licensing process here might be helpful for international readers–if you decide to keep this example.

We've added a bit more detail yet tried to pick out concepts that translate internationally without weighing down the paper with a lengthy description of the US licensing process.

REVIEWER:-Page 13, line 24: how do you define "adaptive licensing" in this context? How is it different from adaptive management built into (or as a condition of) the licensing process/actual license?

We have tried to clarify this in the text.

REVIEWER:-Page 14, line 1: what do you mean by "Robust strategies" in this sentence? This seems vague.

We have clarified this in the text.

REVIEWER:-Page 14, line 20: why wouldn't you place footnote #4 here, upon first mention of the term "boundary organizations"?

We made that change.

REVIEWER:-Figures 2-3: what is the purpose of "*100" and "*10" behind "Alaska" and "Finland" respectively in the legends of these figures?

We have added language to the caption to make this clearer.

REVIEWER:-Figure 5: what does the grey-striated (diagonally-stripped) area of this map mean? It is unclear from the legend and Figure caption.

This definition has been added.

REVIEWER:-Figure 6: could you replace "drawing" in the figure caption with "graph" for clarity?

The authors have changed this.

**Reviewer Two's comments:**

REVIEWER: Comment 1: It is imperative that planners and decision-makers have access to information on uncertainty in not only flows but also in sediment loads so these can be accounted for in the design of new hydropower projects. Change in climate can influence erosion and catchment sediment yield. This change in sediment yield can affect the reservoir storage volume and power production. If this is relevant to the Far North then, I think author should develop some paragraph on this aspect too.

A short statement on this has been added at the end of section 3.1

REVIEWER: Comment 2: In general there are three major sources of uncertainty related to climate change predictions: (1) due to GCM, emission scenarios and the down-scaling method used (2) uncertainty in land use change which is mostly neglected but should be included and (3) uncertainty due to hydrological modeling. These sources of uncertainties make hydrological predictions challenging. Nevertheless quantifying the uncertainty at every step in the modelling process (cascading uncertainty) can address the challenge in quantitative assessment of climate change impacts on catchment hydrology considering the full range of uncertainties involved. Hence what is needed is a careful investigation of sources of uncertainty through cascading and then identification of the major source(s) of uncertainty and focus on reducing that major source(s) of uncertainty. There are recent studies which have pointed out that major source of uncertainty can vary with time and is space dependent. Some discussion on these points could strengthen more Section 3.3 Estimating and reducing uncertainty.

A statement on cascading uncertainty with citations has been added to section 3.3, while trying to avoid lengthening the paper too much more.

[revised manuscript text omitted]